# Transcriptomic and macroevolutionary evidence for phenotypic uncoupling between frog life history phases

Katharina C. Wollenberg Valero[1,†], Joan Garcia-Porta[2], Ariel Rodríguez[3,†], Mónica Arias[4,5,*], Abhijeet Shah[4,6,*], Roger Daniel Randrianiaina[3,7,*], Jason L. Brown[8], Frank Glaw[9], Felix Amat[10], Sven Künzel[11], Dirk Metzler[4], Raphael D. Isokpehi[1] & Miguel Vences[3]

Anuran amphibians undergo major morphological transitions during development, but the contribution of their markedly different life-history phases to macroevolution has rarely been analysed. Here we generate testable predictions for coupling versus uncoupling of phenotypic evolution of tadpole and adult life-history phases, and for the underlying expression of genes related to morphological feature formation. We test these predictions by combining evidence from gene expression in two distantly related frogs, *Xenopus laevis* and *Mantidactylus betsileanus*, with patterns of morphological evolution in the entire radiation of Madagascan mantellid frogs. Genes linked to morphological structure formation are expressed in a highly phase-specific pattern, suggesting uncoupling of phenotypic evolution across life-history phases. This gene expression pattern agrees with uncoupled rates of trait evolution among life-history phases in the mantellids, which we show to have undergone an adaptive radiation. Our results validate a prevalence of uncoupling in the evolution of tadpole and adult phenotypes of frogs.

[1] Department of Natural Sciences, College of Science, Engineering and Mathematics, Bethune-Cookman University, Daytona Beach, Florida 32114, USA. [2] Center for Ecological Research and Forestry Applications (CREAF), Campus of the Autonomous University of Barcelona, 08193 Cerdanyola del Vallès, Spain. [3] Division of Evolutionary Biology, Zoological Institute, Technical University of Braunschweig, Mendelssohnsstr. 4, 38106 Braunschweig, Germany. [4] Ludwig Maximilians University of Munich, Faculty of Biology, Division of Evolutionary Biology, Großhaderner Straße 2, 82152 Planegg-Martinsried, Germany. [5] Institut Systématique, Evolution, Biodiversité, UMR 7205 MNHN-CNRS-EPHE-UPMC-Sorbonne Universités, Muséum National d'Histoire Naturelle, Bâtiment d'Entomologie, CP050, 57 rue Cuvier 75005 Paris, France. [6] Department of Animal Behavior, Bielefeld University, Postfach 100131, 33501 Bielefeld, Germany. [7] Zoologie et Biodiversité Animale, Université d'Antananarivo, B.P. 906, Antananarivo 101, Madagascar. [8] Department of Zoology, Cooperative Wildlife Research Lab, Southern Illinois University, Carbondale, Illinois 62901, USA. [9] Zoologische Staatssammlung München (ZSM-SNSB), Sektion Herpetologie, Münchhausenstraße 21, 81247 München, Germany. [10] Àrea d'Herpetologia (BIBIO), Museu de Granollers-Ciències Naturals, Palaudàries, 102. Jardins Antoni Jonch Cuspinera, Granollers, Catalonia 08402, Spain. [11] Max Planck Institute for Evolutionary Biology, 24306 Plön, Germany. † Present addresses: School of Environmental Sciences, University of Hull, Hull HU6 7RX, UK (K.C.W.V); Institute of Zoology, University of Veterinary Medicine Hannover, Bünteweg 17, 30559 Hannover, Germany (A.R). * These authors contributed equally to this work. Correspondence and requests for materials should be addressed to M.V. (email: m.vences@tu-bs.de).

Many groups of organisms are characterized by distinct life-history phases that strongly differ in morphology and ecology. How different life-history phases evolve, remain stable through time and drive species diversification are major questions in evolutionary biology[1]. In species with complex life cycles, ecological opportunity can present itself to a single life-history phase, or to more than one phase simultaneously. As pointed out by Darwin[2], natural selection may exert wholly different adaptive pressures on larvae than on adults, but selection on traits in either phase may affect the fitness of the other. He suggested that selection, through 'laws of correlation,' affects both phases simultaneously. For example, the lipid-like juvenile hormone in *Drosophila melanogaster* has pleiotropic phenotypic effects both pre- and post metamorphosis[3].

Such linked or coupled evolution of life-history phases, where selection events targeting one life-history phase affect traits in another phase, can best be studied in organisms with discrete phases, such as those with larvae differing considerably from the adult body plan. In addition to holometabolous insects and many parasitic invertebrates, anuran amphibians (frogs) provide an excellent model. Frog life cycles usually include a larval and an adult phase, although many variations on this theme exist, such as direct-development of eggs into froglets, nidicolous (non-feeding) tadpoles or ovoviviparity[4]. A typical tadpole is aquatic and omnivorous/microphagous, with many specific adaptations in the feeding and locomotory apparatus[4–6]. By contrast, adults are typically more terrestrial and strictly carnivorous. Tadpole morphospace evolved particularly fast during the early diversification of frogs[7] and probably played an important role in the adaptive diversification of anurans. Tadpole morphology is phylogenetically informative[8], but the existence of discrete ecomorphological guilds comprised of phylogenetically unrelated species[9] resulting from repeated convergence into similar morphospaces among younger anuran clades[7], suggests extensive homoplasy. Similarly, in adult frogs, many species have striking similarities in ecomorphological traits that can be explained both by evolutionary conservatism and convergence[10].

Harris[11], building on Falconer[12], used data on tadpoles to propose models of evolution of traits coded by the same gene in both phases of the anuran life cycle. According to these models[11] a substantial proportion of traits modified by selection in one phase will be maladaptive in the other, if traits are correlated or coupled in both phases. Coupling of traits coded by the same genes in both larval and adult phases could therefore itself be selected against. This idea suggests an alternative hypothesis of uncoupling, where traits that are partially uncoupled across phases could speed up the rate of specialization in both phases, such that adult and tadpole phases reach different adaptive peaks. But at that point, selection on both phases and on trait coupling itself would stop[11] (Fig. 1). A similar alternative heuristic model that considers uncoupling itself as a trait[11,12] led to a partly similar prediction, that is, uncorrelated traits affecting only one of the phases are fixed at higher frequency than traits correlated across phases.

These models allow us to generate testable predictions for genes associated with external morphological features (Fig. 1). Under coupled evolution of life-history phases in frogs, we would expect expression of the same morphology-related genes in both phases ('phase-pleiotropy'). Modification of these genes would be mostly maladaptive, as it is unlikely that such changes would be simultaneously favoured by selection in both phases. Over time, coupled evolution would be expected to lead to phenotypic changes in larval and adult phases being concentrated on the same clades in a phylogenetic tree. Conversely, under uncoupled evolution of frog life-history phases, either different genes code for morphological traits in each phase, or the same genes do but

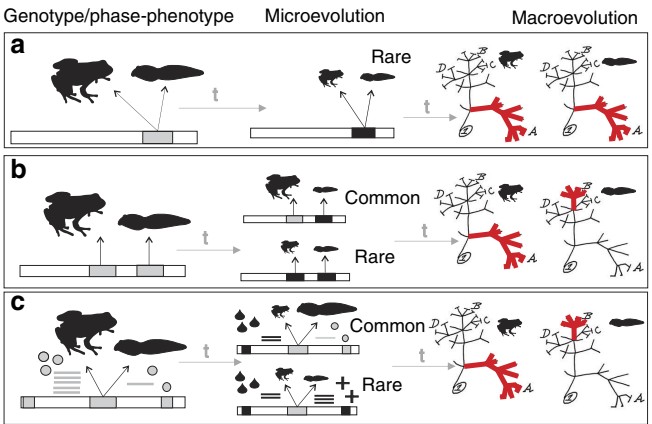

**Figure 1 | Predictions for coupled and uncoupled trait evolution in life-history phases.** (**a**) 'Phase-Pleiotropy' under coupled evolution. The same genes contribute to tadpole and adult morphological features (here: body size). Non-synonymous DNA substitutions are rare due to phase-specific selective regimes. The same genes are equally expressed across phases. Phenotypic evolutionary rates are correlated and rate changes are concentrated on the same parts of the phylogeny (t represents evolutionary time). (**b**) 'Phase-Polygeny' model 1: uncoupled evolution with different genes coding for tadpole and adult phenotype. Non-synonymous DNA substitutions are common due to phase-specific selective regimes. Genes are unequally expressed across phases. Parallel genetic changes are rare, only if favored by selection across phases, and facilitated by linkage. Phenotypic evolutionary rates are not correlated and rate changes are unequally distributed on the phylogeny. (**c**) 'Phase-Polygeny' model 2: uncoupled evolution with regulatory (gene expression) or epistatic changes among tadpole and adult phenotype. The same genes contribute to tadpole and adult phenotype, but are regulated/expressed differently (mRNA— bars, epistatic gene products—tear drops, crosses, grey circles). Independent changes in expression/regulation are common due to phase-specific selective regimes. Transcripts are unequally expressed across phases. Parallel changes in expression/regulation are rare, occurring only if favoured by selection across phases and facilitated by functional interactions between regulators. Phenotypic evolutionary rates are not correlated and rate changes are unequally distributed on the phylogeny.

are differentially regulated ('phase-polygeny'). Differential regulation can evolve from epistatic mutations (for example, in transcription factors), such as the differences in *hox* gene regulation that are known to have caused limb modifications through tetrapod evolution[13]. Under uncoupled evolution, we expect selection to act independently on life-history phase-specific genes, leading to modifications of adult and tadpole phenotypes independently across a phylogeny.

Empirical evidence for either of these hypotheses is elusive. Research has focused on the action of single genes, mostly on those directly associated with the regulation of metamorphosis, but not on genes related to morphological features[14]. Another study reported covariation of adult and tadpole morphology[15] based on convergent phenotypes across frog clades, and hypothesized that similar adaptations result from similar ecological selective pressure on both life-history phases due to the occupation of similar habitats, but not through genetic or functional linkage.

Among the clades studied for covariation between adult and larval traits[15] was the Madagascar-endemic family Mantellidae. With at least 300 species[16], mantellids exhibit an extraordinary morphological and ecological diversity[17]. Adult mantellids cover a wide ecomorphological spectrum including terrestrial, semi-aquatic, arboreal and semi-fossorial habits[17–19]. Likewise, their tadpoles have evolved a high functional diversity including

funnel-mouthed, sucker-mouthed, sand-eating or endotrophic buccal morphologies[19–26]. Mantellids might represent an adaptive radiation as they underwent a rapid early burst of diversification[17] and contain at least one subgroup in which diversification rates decline over time[27]. Given their remarkable diversification within a single geographic region, mantellids constitute an optimal model to test for coupled versus uncoupled evolution of larval and adult morphological features.

Here we combine transcriptomic and macroevolutionary evidence to test for coupled versus uncoupled evolution in life-history phases of frogs. We first test whether genes contributing to frog morphology are expressed evenly or differently across life-history phases of the model organism *Xenopus laevis* and its distant relative, the mantellid *Mantidactylus betsileanus*. We subsequently perform an analysis of phenotypic evolution of life-history phases of the entire mantellid radiation. Our integrative study finds evidence for a high proportion of morphology-related sets of genes expressed differentially between tadpoles and adult frogs, and for uncorrelated phenotypic shifts and evolutionary rates in tadpoles versus adults, in agreement with the hypothesis of uncoupled phenotypic evolution of life-history phases in these amphibians.

## Results

**Uncoupling of morphology-associated genes across phases.** A comparison of all genes with developmental timing of expression compiled in a *Xenopus laevis* database (Xenbase.org; see Methods) revealed that genes associated with morphological feature formation are expressed phase-specifically in tadpoles and adult frogs (Fig. 2; Supplementary Table 1). Of the total number of phase-specifically activated genes, high proportions had gene ontologies associated with morphological features: 9.7% (adult) and 8.1% (tadpole). A far smaller fraction of genes expressed across all phases was morphology-associated (1.6%), as expected under the uncoupling hypothesis.

This pattern of phase-specific gene expression in the model organism, *Xenopus*, was summarized from a diverse array of experiments (see Methods) and we hypothesized it to be conserved among anurans. To confirm this hypothesis for mantellid frogs, we compared whole-body transcriptomes of tadpole and adult *Mantidactylus betsileanus*. The same phenotype-associated gene ontology search terms as in the *Xenopus* analysis were used to mine the *Mantidactylus* transcript annotations (Supplementary Fig. 1). In these transcriptomes, 162 annotated protein-coding genes were distinctly overexpressed in either adult or tadpole phase (Fig. 2; Supplementary Fig. 2). Of these, 54.5% overexpressed genes had morphology-associated gene ontologies in the tadpole, and 24.8% overexpressed genes had morphology-associated gene ontologies in the adult (Fig. 2). These percentages were again higher than that of morphology-associated genes evenly expressed across the two phases (16.3%). Two-sided *Z*-tests found genes associated with morphological feature formation to make up a significantly higher proportion of genes expressed during specific life phases, than of those expressed across all phases, in both *Mantidactylus* and *Xenopus* (Supplementary Table 2).

Some phase-specific genes with morphology-associated gene ontologies were functionally related to one another (Supplementary Table 3); for example, many genes overexpressed in both adult *Xenopus* and adult *Mantidactylus* form an interactome significantly enriched for the KEGG pathway 'osteoclast differentiation' (pathway ID 04380, 7 genes in set, PPI enrichment $P = 1.06e - 5$; FDR 0.00457), which is a key process of formation of the ossified skeleton which in turn generates many characters of the adult phenotype (Fig. 2d;

Supplementary Figs 2 and 3). Among the morphology-associated genes overexpressed in the tadpole of *M. betsileanus* were several keratin protein-coding genes (Fig. 2), and *Xenopus* tadpoles also expressed the keratin gene *XAK-B* (syn. *krt24*), both of which are possibly associated with the formation of the keratinized mouthparts unique to tadpoles. These mouthparts contain important morphological characters frequently used in species identification and relevant to the feeding ecology of tadpoles.

**The mantellid radiation is adaptive.** To understand morphological evolution in life-history phases of the Mantellidae, we first tested whether this clade qualifies as an adaptive radiation. We scored 43 adult and 117 tadpole phenotypic traits as categorical characters for 112 species covering all major mantellid subclades and genera (Supplementary Tables 4 and 5). These data sets were subsequently reduced in dimensionality by non-metric multi-dimensional scaling (MDS) for further analysis, extracting up to four phenotypic MDS variables (Supplementary Fig. 4).

Comparative fitting of six models of species diversification to a molecular mantellid time tree[17] yielded two best-supported models, with 2 units difference of AICc between them (Table 1). The two models agreed in suggesting low levels of extinction, with none at all in model 3 and an extinction rate of $1.23e - 8$ in model 4. Both also were characterized by an exponential decrease in speciation rates through time (Fig. 3). Morphological disparity in adult and tadpole life phases also decreased through time in the first two MDS variables, with negative morphological disparity indices (MDI) of $- 0.355$ and $- 0.014$ (adult phase), and $- 0.106$ and $- 0.089$ (tadpole phase, Fig. 3; Supplementary Fig. 5). The exponential net decrease in diversification rates over time with negligible levels of extinction, as well as the negative MDI values for the most relevant phenotypic summary variables, are consistent with a scenario of adaptive radiation in the family Mantellidae.

**Uncoupling of mantellid phenotypic evolution across phases.** To estimate rates of evolutionary phenotypic change across the mantellid radiation we reconstructed morphological variation based on one MDS variable for each life-history phase on the molecular phylogeny under a Brownian Motion model. Overall rates of phenotypic evolution were significantly higher in the tadpole (mean rate = 0.73) than in the adult phase (mean rate = 0.23, Kruskal–Wallis test, KW-H = 228.89, $P < 0.0001$). We also detected high levels of rate heterogeneity in both adults and tadpoles (Fig. 3). The arboreal genus *Boophis* had a lower than expected rate of phenotypic evolution, especially in adult morphology. Clades with higher rates of change were scattered across the other genera, especially within *Mantidactylus* + *Gephyromantis*, which are ecologically diverse sister genera. Standardized posterior rates of overall phenotypic evolution were not correlated between mantellid life-history phases (Pearson correlation; $r = 0.07$, $P = 0.24$). We found evidence for significantly correlated rates of pairwise change in only 7.5% of all tadpole and adult character comparisons (318 out of 4,255 pairwise comparisons, Supplementary Fig. 6). A timeline diagram of rates of phenotypic evolution showed two peaks, at 8–12 and 20–24 million years ago (Myr ago), at which tadpole rates were distinctly higher than adult rates (Fig. 3), suggesting uncoupling of rate changes during peak phases of morphological evolution.

Referring to macroevolutionary theory[28,29], a phenotypic optimum, also called a 'selective' or 'adaptive' optimum, has been defined as a hypothetical ideal phenotype towards which one or more lineages can evolve[10]. Analysis of selective phenotypic optima based on fitting of Ornstein–Uhlenbeck (OU) models[30] revealed 14 and 13 specific phenotypic optima in

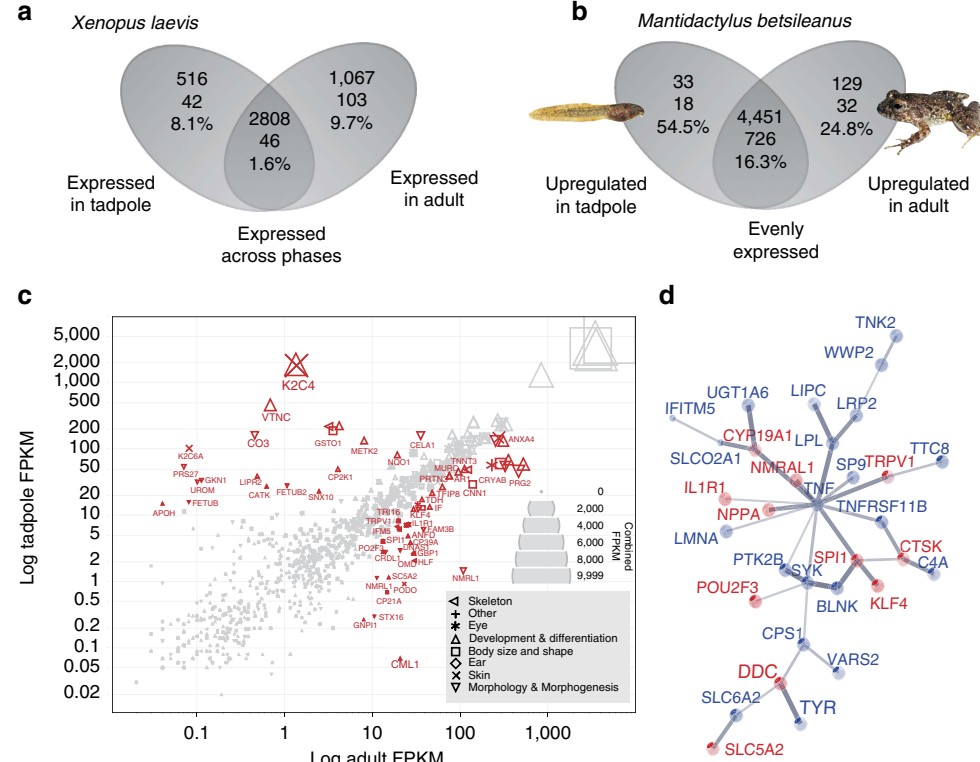

**Figure 2 | Life-history specific gene expression and contribution to phenotype in frogs.** (**a,b**) Venn diagram showing summary statistics of genes differentially expressed between life-history phases tadpole and adult, and combinations of these in (**a**) *Xenopus laevis* (from Xenbase) and (**b**) *Mantidactylus betsileanus* (from two newly sequenced transcriptomes). Numbers in Venn diagrams: top row, number of annotated genes expressed per phase; middle row, number of genes associated with morphological feature formation expressed per phase; bottom row, percentage of morphological-feature-formation-associated genes. (**c**) Morphology-associated gene expression in adult and tadpole *M. betsileanus* shown as scatterplot of log tadpole FPKM (fragments per kilobase of exon per million reads mapped) versus log adult FPKM. Symbols denote different categories of morphology-associated gene ontology terms; if one gene has more than one term the symbols are partly overlapping. Symbol size represents combined adult and tadpole FPKM value, as shown in the stacked FPKM scale (right). Red labelled symbols denote significantly differentially expressed transcripts (protein symbols). (**d**) STRING subnetwork (confidence view) of functional interactions between some genes expressed in adult life phase in *Xenopus* (blue gene symbols) and *M. betsileanus* (red gene symbols). Expressed genes of both species are linked within the same functional network with 'osteoclast differentiation' as the significantly overexpressed function. Line thickness represents strength of confidence for interactions.

**Table 1 | Selection of models for species diversification of mantellid frogs**

| Model | Description | LogL | AICc | ΔAICc |
|---|---|---|---|---|
| Model 3 | *No extinction and exponential variation in speciation rate through time* | *− 893.794* | *1791.64* | *0* |
| Model 4 | *Exponential variation in speciation rate and constant extinction rate* | *− 893.794* | *1793.68* | *2.048* |
| Model 6 | Exponential variation in speciation and extinction rates | − 893.794 | 1795.75 | 4.111 |
| Model 1 | No extinction and constant speciation rate | − 924.282 | 1850.58 | 58.945 |
| Model 2 | Constant speciation and extinction rates | − 924.282 | 1852.61 | 60.976 |
| Model 5 | Constant speciation rate and exponential variation in extinction rate | − 924.282 | 1854.66 | 63.024 |

AICc, Akaike Information Criterion; LogL, Log likelihood; ΔAICc, difference to the AICc.
Columns show for each tested model the values for its LogL, AICc and the ΔAICc of the previous model listed. Best-supported models given in italics.

tadpole and adult life-history phases, respectively (Fig. 3). This involved a respective drop of 133.74 and 286.76 AICc units, in comparison with the model that involved only one optimum across the tree. Most phenotypic optima values were found to be in the range of empirical values (Supplementary Fig. 7). Rates of adaptation towards the different optima were generally high in the adult life-history phase presenting phylogenetic half-lives (the expected time to evolve halfway to an optimum, computed as 'ln(2)/α' ranging from 2.2 to 6.8 Myr, but were highly variable in tadpoles, presenting phylogenetic half-lives ranging from 28 to 0.03 Myr. Overlap of phenotypic optima in adult frogs was extensive, but tadpole morphologies in contrast showed several well-defined phenogroups (Supplementary Fig. 7). Comparing our estimates with 100 replicates of non-convergent OU models did not provide support for convergence in adult or tadpole life-history phases ($P = 0.54$ and $0.15$, respectively; Supplementary Table 6). The analysis implies that 17 shifts between phenotypic selective optima occurred in the tadpole life phase, and 15 in the adult phase (Fig. 3; Supplementary Table 7). The comparison of the topological positions of shift nodes in adult and tadpole phases along the tree revealed that 35% of the shift nodes in tadpole phenotypic optima were at the same nodes as shifts in adult optima. This similarity was higher than that of 1,000 shift randomizations we conducted across the phylogeny ($P = 0.001$).

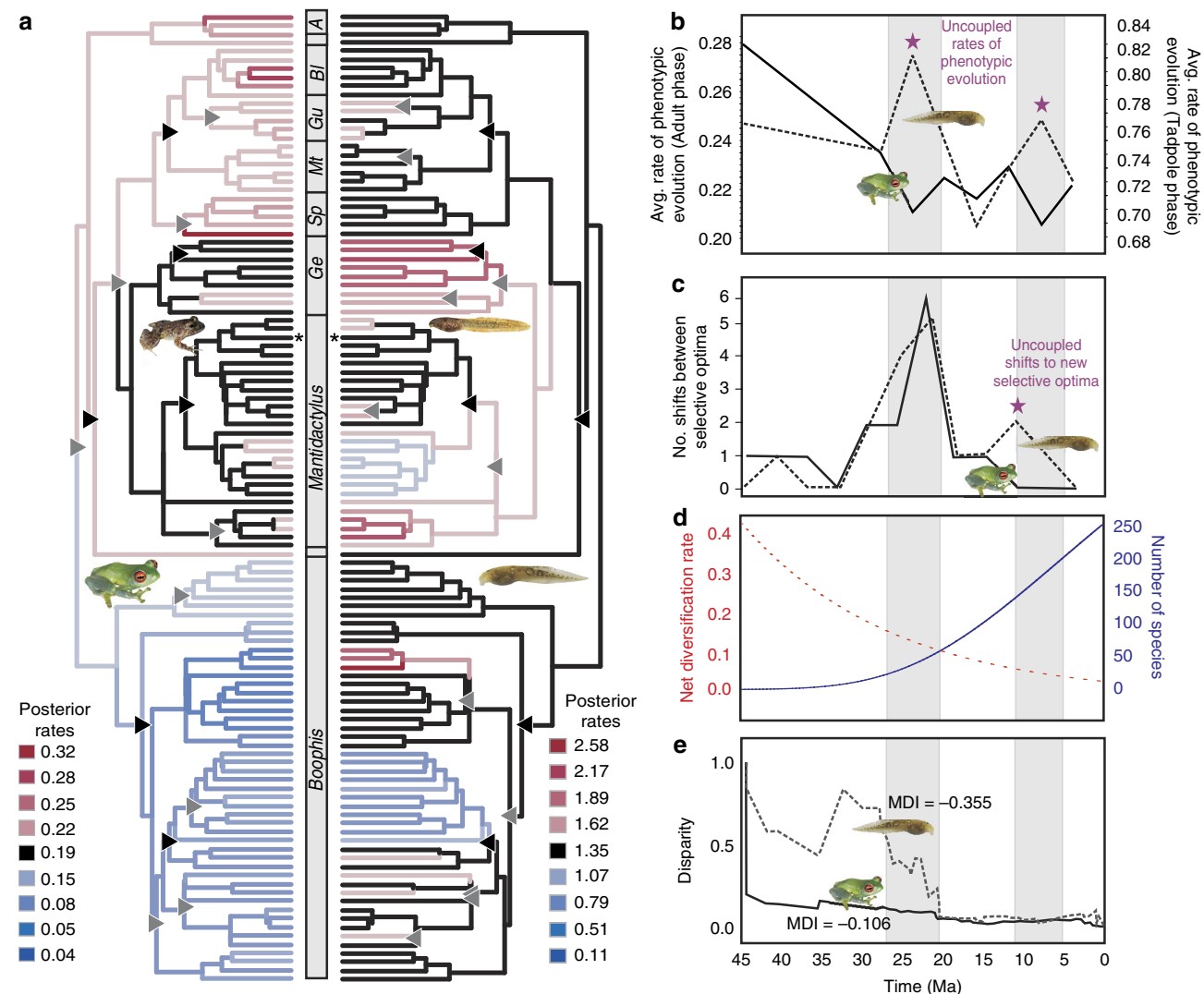

**Figure 3 | Phenotypic optima and uncoupling of character evolution in mantellid frogs. (a)** Phylogeny showing high (red) and low (blue) relative rates of phenotypic evolution in mantellids. Black branches show no rate acceleration or deceleration. Triangles mark shifts between selective optima (black, shared in adult and tadpole phenotypes; grey, unique to either of the two phases). A, *Aglyptodactylus*; Bl, *Blommersia*; Ge, *Gephyromantis*; Gu, *Guibemantis*; Mt, *Mantella*; Sp, *Spinomantis*. The focal species *Mantidactylus betsileanus* is marked with an asterisk. **(b)** Rates of phenotypic evolution through time, showing uncoupled rates at ca. 24 and 7.5 Myr ago (indicated by a star and highlighted by grey bars). The continuous line represents adults, the hatched line tadpoles. **(c)** Plot of the number of shifts between selective optima through time, showing two peaks at ca. 25 and 12.5 Myr ago. The second adaptive peak is characterized by uncoupling of shifts between adults and tadpoles (star). **(d)** Variation of the diversification rate (in orange) and the predicted increase of species richness (in blue) according to the best-supported diversification model found in our study (model 3). This model of diversification assumes an exponential variation of speciation rate with no extinction. **(e)** Disparity through time (DTT) plots visualizing the dynamics of phenotypic diversification in adults and tadpoles of the family Mantellidae, for the first MDS variable. See Supplementary Fig. 5 for full plots including comparisons with simulated curves. Inset photos show representative mantellids.

Shifts of selective phenotypic optima in both life-history phases thus coincided significantly across the phylogeny. The direction of these shifts was, however, not correlated, with only 14% of optima shifts occurring in the respectively same subclade descending from each of these shift nodes ($P = 0.16$). A time diagram showing the number of shifts through time suggests that in both life-history phases, the number of shifts both for adults and tadpoles peaked in the interval between 20 and 24 Myr ago (Fig. 3), whereas a second minor adaptive peak is characterized by shifts in the tadpole phase alone (Fig. 3).

## Discussion

This study combined evidence from phase-specific gene expression in tadpoles and adult frogs with comparisons of phenotypic

diversification in a diverse anuran radiation. We found a significantly higher proportion of morphology-associated genes expressed in specific phases, as compared to the proportion of such genes expressed across all phases, by analysing a comprehensive database of *Xenopus* gene expression. A similar pattern was observed in gene expression levels from adult and tadpole phases of a second frog species, the mantellid *Mantidactylus betsileanus*.

These results indicate that in each life-history phase, a large proportion of morphological features is controlled by a different, phase-specific set of genes. Despite being distantly related, we found that several genes, overexpressed in one phase in *M. betsileanus* and activated in the same phase of *Xenopus*, are functionally related to each other, for instance genes contributing to osteoclast formation and thus associated with skeleton

development. This suggests that phase-specific gene expression patterns leading to morphological feature formation are conserved and comparable among distantly related frog species. The combined results hint at phase-specifically expressed gene sets determining phase-specific morphological feature formation. Hence, gene expression patterns support the hypothesis of uncoupled phenotypic evolution in life-history phases, aligned to models previously defined[11,12] (Fig. 1).

We complemented this evidence derived from gene expression by macroevolutionary analysis of the species-rich mantellid clade to which *M. betsileanus* belongs, based on phase-specific morphological traits. The term adaptive radiation has been used for endemic faunal radiations of Madagascar in previous studies[27,31–34] but evidence for this process, that is, a rapid initial diversification of lineages and rapid filling of morphospace, was found for only a few of these clades[35–37] and was absent in others[38]. In the present study, we support a diversification model for mantellid frogs that combines a drop in speciation rate with low extinction, along with declining morphological disparity indices in adult and tadpole phases. Together with their extensive ecomorphological diversity, this clearly characterizes the Mantellidae as an adaptive radiation.

Both the tadpole and adult life-history phases in mantellids are characterized by distinct functional guilds or eco-phenotypes, for example, 'funnel-mouth tadpoles' or 'leaf-litter frogs'[9,18], and in general, morphological homoplasy is a recurrent pattern in anuran evolution[10]. Yet, our analysis found no evidence for deterministic convergence to similar ecotypes within mantellids. This indicates either that there is not enough phylogenetic separation between morphologically similar phenotypes to be considered convergent, or that their superficial similarities found in other studies are not reflected by identical states in the characters assessed herein. From our analyses, though, it is obvious that subclades of Mantellidae differ in the rate and pattern of phenotypic evolution, for instance with a slower adult morphological evolution in clades of *Boophis* which have a uniform 'tree-frog' adult phenotype and arboreal habits. This evidence for adaptive phenotypic diversification of the measured morphological characters justifies the use of these metrics to test the hypothesis of uncoupling of selected phenotypes within mantellids.

Our data show that mantellid diversification was accompanied by novel ecomorphological adaptations in tadpole or adult phases, but not simultaneously in both phases. Although the position of shift nodes in tadpole and adult phases was more often concordant than expected by chance, the direction of the shifts differed because in one branch resulting from the respective node it was the adult phase to shift into a new phenotypic optimum, while in the other branch, the tadpole phase shifted. Despite this lack of phylogenetic concordance, we found evolutionary transition into new phenotypic optima in tadpole and adult phases to cluster temporally (Fig. 3c), which at first glance does not seem to be concordant with the uncoupling hypothesis. A major peak of shifts into new phenotypic optima in both phases was found early in the mantellid radiation, between 20–24 Myr ago. This temporal concordance among phenotypic shifts in the two phases might be a simple stochastic effect because in the earlier stages of the radiation, the mantellid tree contains only a limited number of nodes on which phenotypic shifts could occur. The peak of shifts coincides with an intensifying of the Monsoon season in Madagascar at the upper Oligocene/early Miocene boundary[39–42], and it warrants further analysis whether this environmental change could have simultaneously induced a change in both adult and tadpole selective regimes, or whether the 20–24 Myr ago peak can be explained with an alternative scenario.

The combined evidence from phylogenetic occurrence of phenotypic shifts and character evolution analysis supports the idea that morphological change through evolution is largely uncoupled between life-history phases. Most of the descendant clades of shift nodes that diversify into novel optima show phenotypic diversification either in the tadpole or the adult phase, and only 14% of all clades concurrently change in both life-history phases. Furthermore, pairwise phylogenetically corrected comparisons among pairs of characters show that 92.5% of all pairwise character comparisons are not correlated across the tree.

Rates of phenotypic evolution were uncorrelated among the two life-history phases (Fig. 3b), which provides additional support for the uncoupling hypothesis. In adults, the rate of phenotypic change is initially high but then decreases and reaches a relatively steady level around 25 Myr ago. In contrast, the tadpole phase shows strong peaks of phenotypic rate change in two time intervals, roughly between 25–20 and 15–10 Myr ago. Along with the overall higher rates of phenotypic evolution in tadpoles, this identifies the larval stage as the main driver of the mantellid radiation; a hypothesis corresponding well with earlier reports of rapid diversification in tadpole morphospace observed during early anuran diversification[7].

In conclusion, our study uncovers support for macroevolutionary uncoupling of phenotypes among life-history phases of a major, endemic and adaptive frog radiation. A member of this radiation, and another distantly related frog species both show uncoupling in morphological feature formation-related gene expression among life phases. This support for the uncoupling hypothesis from independent lines of evidence, while unprecedented to date, was not unexpected; it allows for a mechanism to generate and maintain such strikingly different life-history phases in the first place. While the original conceptual model of Harris[11] only refers to one gene coding for one trait, we have shown here that summary variables of adult and larval phenotypes based on comprehensive trait data sets also evolve at uncorrelated rates in tadpoles and adult frogs, validating the original model's predictions. Furthermore, morphology-related gene use differed between phases, and so did gene expression levels. Since most direct genotype-phenotype connections in anurans are unknown, and several of the traits we studied might be either genetically, or functionally linked to each other, we here flag the study of the exact degree of coupling between linked and unlinked traits in organisms with complex life cycles, and the underlying genomic architecture, for further study. The morphologically unique tadpole phase is already known from clades splitting off the earliest nodes of the anuran tree and their fossil record dates back until the Early Cretaceous[43]. As a testable hypothesis, we predict that the degree of uncoupling, both in terms of gene expression and phenotypic evolution, is more pronounced in anurans than in salamanders whose larval phase is ecologically and phenotypically more similar to their adult phase.

## Methods

**Gene expression in *Xenopus*.** Data on gene expression across *Xenopus laevis* developmental phases were downloaded from Xenbase.org[44]. The 167,519 Xenbase entries were genes expressed per specific experiment. A portion of 2,130 entries of the database was unannotated. A total of 120,381 entries were sourced from cDNA Library/EST experiments; 1,500 entries were sourced from immunohistochemistry experiments, and 45,638 entries from *in situ* hybridization experiments (see Supplementary Methods for additional information on the Xenbase entries used). We mined this database for records of gene expression across all phases of development, and for phase-specific gene expression of tadpole (defined as stage NF9-44) and adult stage of development. Entries were annotated for their expression in stages of development. Genes were binary-coded based on their expression profiles in different life-history phases (only tadpole, only adult and combinations of these), and counted for summary statistics. Groups of phase-specifically expressed genes, and genes expressed across phases, were assigned to Gene Ontologies using gprofiler[45,46] with *Xenopus tropicalis* (Ensembl)[47] as

reference organism. Gene Ontologies were scanned for GOs related to morphological feature formation and counted for summary statistics. Genes per life-history phase that were associated with morphological feature formation are listed in Supplementary Table 1.

**RNAseq and gene expression in *Mantidactylus*.** A stage 27 tadpole and an adult individual of *Mantidactylus betsileanus* were killed with anaesthesia and subsequent overdose of MS222, and dissected immediately afterwards. Samples of all tissues excluding the gut were preserved in RNAlater and frozen at $-80\,^{\circ}$C. After standard RNA extraction, RNA was prepared for sequencing following the Illumina TruSeq mRNA protocol and sequenced on an Illumina NextSeq machine. A total of 130,607 transcripts were assembled from the pooled reads of both samples, and read counts per sample were normalized by transforming them into FPKM (fragments per kilobase of exon per million reads mapped[48]. In all, 4451 ORF-containing transcripts were annotated to gene symbols and gene functions (using Gene Ontology and KEGG taxonomies[49,50]) using Blast2GO[51] and TRINOTATE[52], either for gene symbol (via blastx) or gene function (via InterPro domain searches[53]). We identified 641 differentially expressed transcripts using edgeR in Bioconductor[54] (with dispersion value = 0.4). The true number of differentially expressed genes among phases might be higher or lower, as our sample size of *Mantidactylus* was not suited to uncover biological variability in transcript numbers - both the numbers of evenly and of differentially expressed genes have to be considered as random samples of a true number. We repeated the analysis with a method suited for small sample sizes (NOISeq in Bioconductor[54,55]), which recovered 562 outlier genes also included among the 641 genes obtained from edgeR. We used this more conservative estimate for further analysis. Genes considered to be unevenly expressed among tadpole and adult phases had FPKM ratios between adult and tadpole transcripts of $>1.6$ and $<0.4$, respectively.

The list of phenotype and morphological feature-associated GO search terms from the *Xenopus* analysis was used to text mine all *Mantidactylus* annotations using a Perl script (Supplementary Data 1). A complete list of search terms and their frequency of occurrence in the *Mantidactylus* transcriptomes is given in Supplementary Fig. 1. Visualization and analysis of the morphological feature-associated data set was performed in TABLEAU[56]. The ratios of morphological feature-associated genes that were overexpressed in the tadpole phase and the adult phase, and those that were evenly expressed across both phases, were subjected to summary statistical analysis and $Z$-tests to determine differences in proportion between morphology-associated genes within phases and genes expressed in all phases. Two-sided $Z$-tests found genes associated with morphological feature formation to be in significantly higher proportion in genes expressed during specific life-history phases, than in genes expressed across all phases in both *Mantidactylus* and *Xenopus* (Supplementary Table 2).

To infer whether phase-specifically expressed, morphological feature-associated genes in *Xenopus* and *Mantidactylus* were functionally connected with each other, the lists of genes from each phase (genes expressed in tadpole phases and genes expressed in adult phases) for both species were submitted to the STRING database[57], and interaction networks were generated for both phases (using human as reference organism). Functional connection and term overrepresentations were quantified in STRING, and subnetworks that functionally connected genes from each life-history phase between both species are given in Supplementary Table 3.

**Scoring morphological characters in mantellids.** To study the macroevolutionary patterns of adult and tadpole phenotypes through the mantellid radiation, we compiled a morphological dataset of 43 characters for adults of 231 species and candidate species, based both on data from the literature and extensive personal examination of voucher specimens by the authors (see Supplementary Methods for more detail). For larvae, a dataset of 117 characters for 170 mantellid species was assembled for this study, based on tadpole voucher specimens that had previously been identified by DNA barcoding[25]. Data sets of adult and tadpole measurements overlapped and were jointly analysed for 113 mantellid species. Mensural characters were transformed into categorical character states for downstream analysis. Multistate characters implying a continuum were coded as ordered. Different coding schemes (binary character states only, all characters unordered, character jackknifing) were explored to verify that the different number of characters in the tadpole and adult data sets, and the coding scheme itself, had no influence on our main findings (Supplementary Tables 8–10). See Supplementary Tables 4 and 5 for lists of all characters and character states and Supplementary Data 2 and 3 for the respective Nexus files for analysis.

To create a single metric from all larval and adult morphology characters for subsequent analyses, we calculated the number of character state transformations between all pairs of species using PAUP* 4.0 (ref. 58). The morphology distance matrix was then reduced into a new set of variables using the ordination method of dissimilarity multidimensional scaling (MDS, in SPSS 15.0). For analyses of phenotypic optima, we extracted four MDS variables, the first two of which contained a significant portion of overall variance (low stress values $>0.2$), while variation was maximized on a single MDS variable per phase for the comparison of phenotypic evolutionary rates. MDS calculations elucidate the structure underlying a multivariate data set by providing a geometric representation. We are aware that in our morphological data set, some variables might be autocorrelated, and that the

used MDS procedure does not remove this autocorrelation, so that specific characters might be overweighed if they are reflected in more than one of the variables. However, it is inherently difficult to distinguish between (a) autocorrelation due to direct ontogenetic influences (such as tail height and tail muscle height, or forelimb length and hind limb length), and (b) autocorrelation that arose by homoplasy through similar selective pressures acting simultaneously on different parts of the body. We therefore considered this method (which will emphasize morphological distinctness among different morphological clusters of frogs and tadpoles) as most appropriate to understand possible correlations among adult and larval morphologies and phenotypic evolution of life-history phases.

**Analysis of macroevolutionary patterns in mantellids.** All analyses in this study are based on a previously published time-calibrated phylogeny of mantellids with near-complete taxon sampling[17]. This phylogeny was calculated from DNA sequences of three mitochondrial gene segments, and constrained to a backbone tree generated from multiple mitochondrial and nuclear markers. As morphological data were not available from all species and candidate species (especially for tadpoles) we pruned any taxa lacking the respective morphological data from the tree. To infer whether the mantellid radiation shows signatures of an adaptive diversification process, we tested for an 'early burst' pattern in species diversification, consisting of high diversification rates at the beginning of the radiation followed by rate deceleration towards the present[59]. We compared the second order Akaike Information Criterion (AICc) for six alternative models of species diversification[60] using the package *RPANDA*[61] in R[62]. We then used *auteur*[63] in R to model rate heterogeneity on a phylogeny (assuming a Brownian motion model) using a reversible-jump MCMC algorithm. We ran two chains, each for one million generations, discarding the first half as burn-in. Rates of character evolution were calculated along the mantellid guide tree, and estimated rates of character evolution were then visualized as the geometric mean of rates sampled in each Markov generation, colouring in red versus blue the branches with accelerated or decelerated phenotypic evolution as compared to the median.

We furthermore calculated the morphological disparity of subclades through time with the packages *Ape*[64] and *Geiger*[65] in R, for each of the MDS variables. This analysis plots disparity for observed and simulated data against node age and calculates a MDI, thus quantifying the difference in relative disparity of a clade to that expected under a Brownian motion model[66]. If MDI values are negative, this indicates that subclades have lower disparity than expected by Brownian motion, which often characterizes adaptive radiations.

Adaptive phenotypic optima and selective regimes were detected by implementing different Ornstein–Uhlenbeck (OU) models using the package *surface*[30]. Once the different selective regimes were defined for both adult and tadpole life-history phases, these were visualized on the phylogeny. For both adult and tadpole phases, we compared the extent of convergence in phenotypic optima obtained to a null distribution using 100 replicates of non-convergent OU models. To understand the degree of coupled versus uncoupled evolution between larval and adult life phase morphology, we first searched for correlated evolution among single characters of adults and tadpoles using the Multistate option in BayesTraits[67] (see Supplementary Methods for more details). This analysis, fitting OU models with *surface* to the guide tree in both adults and tadpoles, allowed us to subsequently explore covariation between the shifts to novel phenotypic optima experienced by adults and tadpoles, that is, whether (1) shifts occurred at the same nodes (shift nodes) in both life-history phases, and whether (2) the descendant clade with the new optimum following such shift nodes was the same in both life phases. We counted the number of shift nodes that were common to both adults and tadpoles and compared this estimate with a null model. For this we divided the number of shift nodes shared in both adults and tadpoles by the total number of shifts in tadpoles (the life-history phase with the highest number of shift nodes, see the Results). We then randomized the total number of shifts found in adults and tadpoles 1,000 times (independently) across the phylogeny, each time recalculating the above-described ratio. We then compared the distribution of simulated ratios with our empirical ratio to estimate how often the ratio derived from randomizations was equal or higher than the empirical ratio. Deviation of the direction of shifts in phenotypic optima from a null model was tested using the same procedure, but limited to nodes whose descendant sublineages changed their selective optima simultaneously in adult and tadpole life phases. For example, a split into sublineages A and B, is unidirectional if changes in selective optima occur in lineage A and / or B both in adult and tadpole life phases. It is not if changes occur in lineage A in one life phase, and in lineage B in the other phase. The number of shifts among phenotypic optima, and average rates of phenotypic diversification obtained from the *auteur* analysis, were then partitioned in intervals of 4 Myr ago (from tip to root of the phylogeny) and plotted against time, to visualize temporal correlation between adult and tadpole phenotypic evolution. Interval values were correlated to infer covariation of adult and tadpole shifts in selective optima and rates of trait evolution.

**Ethical approval.** All procedures performed involving animals were in accordance with the ethical standards of the institution or practice at which the studies were conducted. Collection in and export of materials from Madagascar were performed with all necessary permits of the Malagasy authorities.

**Data availability.** Data that support the findings of this study (RNAseq raw reads) have been deposited in the NCBI Short Read Archive (SRA) with the accession code (Bioproject) PRJNA357636. The authors declare that all the other data supporting the findings of this study (lists of morphological character states and state matrices) are available within the paper and its supplementary information files.

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

## Acknowledgements

We are grateful to the authorities of the Republic of Madagascar for research, collection, and export permits, and to M. Scherz for copyediting of the manuscript. This study was supported by a grant of the Deutsche Forschungsgemeinschaft to M.V. (VE247/2-1), a DAAD fellowship to R.D.R., a Georg Forster fellowship of the Humboldt Foundation to A.R., and HBCU-UP NSF grant 1435186 to R.D.I. and K.C.W.V.

## Author contributions

R.D.R., F.G., S.K. contributed to the data generation. M.A., J.G-P. and F.A. contributed to data generation and data analysis. A.R., A.S. J.L.B., D.M., R.D.I. contributed to the data analysis. K.C.W.V. and M.V. conceived and designed the study, contributed to the data analysis and wrote the manuscript. All authors contributed to the writing and revision of the manuscript.

## Additional information

**Competing interests:** The authors declare no competing financial interests.

**Publisher's note**: 

