## [Peer Review File · Nature Communications]

Reviewers' comments:

Reviewer #1 (Remarks to the Author):

Wollenberg et al. present a thoughtful, well-analyzed study of uncoupled phenotypic evolution in frog life history stages. Their interest is the contribution of different life history stages to the macroevolution of lineages. They explore phase-specific gene expression in *Xenopus laevis* and *Mantella betsileanus*, thus comparing distantly related taxa in order to infer generalizations, and using the mantellid clade to consider the evolution of an adaptive radiation. They found peaks of gene expression that were present specifically in the tadpole and adult stages, and some in both. Because of the disjunction of presence, and the focus on morphological traits (highly appropriate), they found good evidence for uncoupled phenotypic evolution in the two life history phases examined. As they indicate in their discussion, this is not surprising (and actually long inferred); however, it is most useful to now have available a large body of genetic data, a careful analysis, and three bodies of evidence for a non-uniform pattern of evolution of tadpoles and adults. I like the introduction of such terms as "phase-pleiotropy" and "phase-polygeny." The methods used for the analysis are appropriate and up-to-date. It is an impressive study of potentially broad interest, and as the authors indicate, should promote new avenues of research.

I do have some quibbles, and urge the authors to clarify some aspects in order to further substantiate their work.

First, they present information for three life history phases in *Xenopus* (tadpole, metamorph, and adult), and two in *M. betsileanus* (tadpole and adult). Obviously, mantellid tadpoles metamorphose; why were only tadpole and adult stages recognized for this study? And were metamorphs included as tadpoles, or as adults, in the analysis? This makes comparison with *Xenopus* ineffective, and the authors do not explain this at all. Also, did the analysis take into consideration lengths (absolute and proportional) of the phases in the taxa? This might influence the numbers of peaks, etc.

Second, the first sentence of the Abstract should be re-written. To say "Anuran amphibians have the most derived biphasic life cycle among tetrapods" implies that all tetrapods have biphasic life cycles, which of course is not what they mean, and it doesn't allow for the fact that some anurans are considered not to have biphasic life cycles. They should state clearly in the introduction that they are defining (apparently) "biphasic" as having tadpoles and adults, whether the tadpoles are free-living (the usual definition of biphasic) or not (i. e. direct-developing terrestrial forms in which the tadpoles hatch fully metamorphosed juveniles, or live-bearing forms in which the tadpoles are maintained in or on the body of a parent, often through metamorphosis such that juveniles are "born."

Third, one need not invoke an environmental change scenario (line 285 et seq.) to explain the temporal shifts; the differences in habitat preferences as the adaptive radiation proceeds is sufficient. Suggest either proposing several scenarios (succinctly) or some other approach.

Fourth, I don't think it is at all surprising that a lot of the phase-specific gene expression is conserved, given that it has to do with, for example, osteoclast appearance and bone development—such examples should be temporally conserved across vertebrates (with consideration of cephalization, and other such phenomena).

There are several wordings and other grammatical usages that might be improved for readability, e. g. "were found" rather than "turned out" (line 45), "composed of" rather than "joining" (line 79), "such that" rather than "until" (line 88), and many others. And "tadpoles did NOT serve" Harris anything (line 82); he (and Falconer) used data on tadpoles... There are also some word-order issues, but a quick copy-edit will take care of all of these minor points, if the authors wish.

All in all, very nice work!

Reviewer #2 (Remarks to the Author):

Writing: drop all the directional words: though, indeed, likewise, however, etc. etc. Let the reader decide! - - I dislike the use of parenthetical words placed within a sentence! - - some wording needs attention = "phases are in Mantellidae" - the whole manuscript would benefit from a strong edit - -

Subject: I could not follow all the details of your analyses, but I found the results and discussion very exciting. The process you discuss is probably wide spread in the frog/tadpole sphere, and I hope you pursue other avenues: other taxa, phenotypic plasticity, other reproductive modes, etc.

Reviewer #3 (Remarks to the Author):

Wollenberg Valero et al. present a well designed and well conducted study on the independence of morphological evolution in tadpoles and their adult stage in frogs. The paper analyzes the (mostly) independent morphological evolution of larvae and adults in a substantial sample of Madagascan mantellid frogs, based on transgenomic and morphological data. An early attempt addressing adult morphology in comparison to larval morphological diversity in frogs was published by Bossuyt and Milinkovitch (2000), but the present study goes far beyond. To the best of my knowledge no other study in amphibians has addressed the question of independent larval/adult evolution in frogs in this depth and clarity. I am confident that this work will not only appeal to frog people. Researchers working on animals with complex life cycles (e.g., insects) in general will be interested to read this work. Furthermore, it has relevance to other ongoing projects that try to understand the evolution of Madagascan fauna. In sum, I think this is an important paper that will have an influence on the thinking in the field.

Wollenberg Valero et al. presented a large dataset, including transgenomic data of two taxa and morphological data for more than 100 species. Their hypothesis building and argumentation is clear and the evidence supports the conclusions strongly. The

supplementary materials cover all aspects of the analysis and all methods applied in reasonable depth. The figures provided are well done and give sufficient (and necessary) visual support to the ideas expressed in writing. The text is well structured.

Reading the text, I had only a few minor issues and I would like to share these thoughts, so the authors may consider them for improving this work:

I am not sure if the current title is the best choice. Anyone working on animals with complex life-cycle will actually expect that the life history phases ARE uncoupled, otherwise it would not make much sense to have a complex life cycle in the first place. The authors acknowledge that fact themselves (lines 321-323). In other words, the title seems to highlight the obvious rather than the unexpected new. The merits of this work are, in my opinion, in the combination of transgenomics and adaptive radiation/phenotypic evolution analyses in order to present sound and strong evidence for the common assumption of uncoupling. Maybe the authors are willing to reconsider that title and include some more key words referring to the major cornerstones of this work. It is a matter of taste, but I think the current title does not quite nail it.

Technically sound data: I need to point out to the editors that I am familiar with some but not all methods applied in this study. Generally, I have the impression that the explanations in the Supplements have an appropriate level of explanation to offer the expert reader good depth of detail to reconstruct what had been done in this work. That said, some minor questions arose during my reading:

The morphological characters are presented in tables. The quality of character documentation is very mixed. Although I have plenty of experience in tadpoles I easily bumped into character definitions that I simply do not understand (for example, 117: Lateral space?; 116: classify all possible color patterns in only three states?; what were the landmark point for measurements; what is "body length axis" , etc., etc.). Obviously character definition in morphology can be very difficult if meant to be reproducible. Certainly, this submission is not necessarily the place to present a lengthy documentation of morphological character states to exclude all ambiguity. Therefore, I am NOT asking for a revision of those tables(!), however, one question might be in place: Given the ambiguity or fuzzy definition of some morphological characters, does this ambiguity hold the potential to change the conclusions of the paper if morphological characters were handled differently (for example, five subjective color pattern [116] states rather than three?). I'd like to see the authors consider that and comment on it with a line in an appropriate section of the text.

The morphological datasets were subject to maximum parsimony analyses "to understand how phylogenetically informative the morphological data are" (Supplement page 3). The connection between the morphological parsimony analysis (line 178; Figure S4) to the subsequent steps of analyses have not become fully clear to me, because there is also the molecular tree (Wollenberg et al 2011) that was used as backbone, for example, to reconstruct morphological variation under a Brownian Motion model to determine rates of phenotypic evolution... In other words: There is the Wollenberg-2011 backbone molecular tree and there is the morphological parsimony analysis and the resulting morphology based

tree (Fig. S4) and I am not sure why the latter is needed. Does the morphological parsimony tree feed directly into any of the subsequent analyses? Either it is obsolete, or it is essential and I missed some connections. A clarification will be welcome.

If the parsimony analysis of morphological features directly feeds into subsequent analysis, then more questions follow. The morphological datasets contained phylogenetic signal (line 180). Yet, there are one has to ask about the process of transforming the many morphometric, quantitative and potentially continuous character states into categorial [sic.] (better: discrete). In most characters the metric intervals between character states were set equal, in others the intervals of values are unequal. I could not find a rationale for these different data conversions. It could be that the prior knowledge (bias?) of the authors may have shaped the decision of where to set the boundaries. Furthermore, the authors preferred coding of many characters as multi-state characters (State 0: absent, State 1: X; State 2: Y; for example, adult Char 1) that could alternatively be coded as two characters (absent/present; X/Y). These two ways of coding influence the reconstruction of apomorphies at nodes. Presence of a structure (in the example, presence of the femoral gland as a organ) cannot be reconstructed as synapomorphy at any node if coded as multi-state character (only the two subforms of the femoral glands). I wonder if these two different ways of handling character coding would have any effect on subsequent analyses of phenotypic evolution.

Down-weighting characters that are variable intra-specifically or may be affected by preservation artifacts may sound reasonable at first. However, what is the justification of the amount of down-weighting? The authors cite Goloboff et al. (2008) and mention "upweight % 8 downweight 4%" in the supplements. Why were these figures chosen and not others? Is there a justification for these figures, why not more/less? As a reference point and for transparency: would it change the results and conclusions if all characters were weighted equally?

I am not familiar with the concept of "selective phenotypic optima". When it is first mentioned in the text [line 212] no explanation or citation is given. The explanation in the Supplement section (page 4) did not quite clarify it for me. Sorry for my ignorance, but with respect to the broad readership and the frequent use of phenotypic optima in the text and central role of the term/concept, this work would benefit from making the concept more accessible for the reader.

Recommendation: Accept with revisions

A. Haas

Author responses to reviewer comments

We are grateful to the reviewers for the positive evaluation for our manuscript. They provided a series of valuable comments that hinted at several omissions and manuscript sections in need of improvement. We found these suggestions very constructive and have attempted to revise the manuscript accordingly. In the following we reproduce the original reviewer comments, and provide a point-by-point response to each of them.

Reviewer #1 (Remarks to the Author):

Wollenberg et al. present a thoughtful, well-analyzed study of uncoupled phenotypic evolution in frog life history stages. Their interest is the contribution of different life history stages to the macroevolution of lineages. They explore phase-specific gene expression in *Xenopus laevis* and *Mantella betsileanus*, thus comparing distantly related taxa in order to infer generalizations, and using the mantellid clade to consider the evolution of an adaptive radiation. They found peaks of gene expression that were present specifically in the tadpole and adult stages, and some in both. Because of the disjunction of presence, and the focus on morphological traits (highly appropriate), they found good evidence for uncoupled phenotypic evolution in the two life history phases examined. As they indicate in their discussion, this is not surprising (and actually long inferred); however, it is most useful to now have available a large body of genetic data, a careful analysis, and three bodies of evidence for a non-uniform pattern of evolution of tadpoles and adults. I like the introduction of such terms as "phase-pleiotropy" and "phase-polygeny." The methods used for the analysis are appropriate and up-to-date. It is an impressive study of potentially broad interest, and as the authors indicate, should promote new avenues of research.

I do have some quibbles, and urge the authors to clarify some aspects in order to further substantiate their work.

Response: Thank you for these encouraging comments. In particular, we are glad that the terms we are using (e.g., "phase pleiotropy" and "phase polygeny") have been well received. We reproduced them more prominently in the legend of Figure 1 to increase visibility. Below we provide more detailed responses regarding the specific quibbles.

First they present information for three life history phases in *Xenopus* (tadpole, metamorph, and adult), and two in *M. betsileanus* (tadpole and adult). Obviously, mantellid tadpoles metamorphose; why were only tadpole and adult stages recognized for this study? And were metamorphs included as tadpoles, or as adults, in the analysis? This makes comparison with *Xenopus* ineffective, and the authors do not explain this at all. Also, did the analysis take into consideration lengths (absolute and proportional) of the phases in the taxa? This might influence the numbers of peaks, etc.

Response: Having a complete time series of gene expression data sets in multiple species would be optimal, but due to the high anticipated cost of such an undertaking this would ideally be subject to future grant proposals instead. Adding such datasets would be useful for revealing the exact onset/offset of each expressed gene, but beyond adding more details would be unlikely to convey the same intellectual advance for the field that our present study already delivers with its "snapshots" of gene expression from two species.

The rationale for including the metamorph phase of *Xenopus* in statistical comparisons of number of phase-specifically expressed genes, was to provide an additional data point for phase-specific gene expression associated with morphological structures. This data point is independent from any *Mantidactylus* data since the Z-test did not involve a comparison between the two species. Instead, the three phases in *Xenopus* were compared separately against genes expressed across all phases in *Xenopus* with the Z-test, and the two investigated phases of *Mantidactylus* were compared separately against genes expressed across all phases in *Mantidactylus* with a separate Z-test (Figure 2A, Supplementary Table S2).

For the overexpressed genes per phase that are common between *Mantidactylus* and *Xenopus* (shown as red/blue gene network in Figure 2), we included only adult phase-specifically expressed genes from both species (pre-metamorphosis), and in another analysis shown in the Supplementary Materials we included only tadpole-specifically expressed genes from both species (post-metamorphosis), see Supplementary Table S3, and Supplementary Figures S2 and S3. The *Xenopus* metamorph data was not included in any of these comparisons either. We clarified this now better in the figure legends of Supplementary Figures S2 and S3.

Acknowledging the concern of the reviewer, and to simplify the message of our manuscript, we have decided to remove the information on the metamorph-specifically expressed genes from the manuscript, and only strictly include tadpole and adult phases for both species in the paper.

Second, the first sentence of the Abstract should be re-written. To say “Anuran amphibians have the most derived biphasic life cycle among tetrapods” implies that all tetrapods have biphasic life cycles, which of course is not what they mean, and it doesn’t allow for the fact that some anurans are considered not to have biphasic life cycles. They should state clearly in the introduction that they are defining (apparently) “biphasic” as having tadpoles and adults, whether the tadpoles are free-living (the usual definition of biphasic) or not (i. e. direct-developing terrestrial forms in which the tadpoles hatch fully metamorphosed juveniles, or live-bearing forms in which the tadpoles are maintained in or on the body of a parent, often through metamorphosis such that juveniles are “born”).

Response: We agree with the reviewer that the use of the term “biphasic” was an oversimplification. We have replaced it with “complex life cycle” in all mentions in the text remaining after the other corrections we made (see below).

The first sentence of the abstract has been rewritten to read “Anuran amphibians undergo striking, phased morphological transitions during development, but the contribution of such different life history phases to their macroevolution has rarely been analyzed. .

We added a sentence in the introduction to reflect variations of the complex life cycle: “...Frog life cycles usually include a larval and an adult phase, but variations of this theme include direct-developing tadpoles, nidicolous tadpoles, or ovoviviparity (Altig & McDiarmid 1999)...”

Third, one need not invoke an environmental change scenario (line 285 et seq.) to explain the temporal shifts; the differences in habitat preferences as the adaptive radiation proceeds is sufficient. Suggest either proposing several scenarios (succinctly) or some other approach.

Response: Thank you for this comment. We totally agree with this point, which includes two aspects:

First, the occurrence of phenotypic shifts in itself does not require any environmental change scenario; such shifts certainly can and will occur during radiation into an existing environment. In our text however we did not mean to refer to the shifts themselves, but to the temporal coincidence of adult and tadpole shifts, and clearly, we did not phrase this adequately. We have rewritten and restructured the entire paragraph to make this distinction (occurrence of shifts vs. temporal co-occurrence of shifts in both phases) clearer.

Second, although it is of interest that the temporal clustering of shifts coincides with a time of environmental change, this is simply a correlative coincidence, and there is no evidence for a causal effect. Our previous text therefore included overstatements by hypothesizing such a causal effect. We have now rephrased this by (a) mentioning the possible stochastic effect first (temporal shifts concentrate in early phases of the radiation when only few nodes exist) and (b) mentioning the fact that the concentrated shifts occur in the same period as environmental change merely in a correlative phrasing, stating that a possible causal relationship would require further study.

Fourth, I don’t think it is at all surprising that a lot of the phase-specific gene expression is conserved, given that it has to do with, for example, osteoclast appearance and bone development—such examples should be temporally conserved across vertebrates (with consideration of cephalization, and other such phenomena).

Response: We agree that this outcome is not completely surprising. The main point of our study is that different genes are involved in forming morphological structures in adults vs. tadpoles, and by showing that these processes are conserved (*Xenopus* vs. *Mantidactylus*) we provide a means to link the extensive *Xenopus* data to the macroevolutionary analysis of the species-rich group of mantellids (given the small number of species and conserved morphology of *Xenopus* and relatives, our macroevolutionary approach could not have been applied to these frogs). Yet, it is surprising is maybe that so little empirical evidence has so far become available (as evidenced from the paucity of studies about uncoupling in the literature) showing that developmental gene expression patterns are conserved among different groups of vertebrates, and that evolution of both phases is uncoupled.

We have attempted to make the connection between the *Xenopus* and *Mantidactylus* data clearer by rephrasing a few sentences at the beginning of the Results section, and have also mentioned more clear in this same paragraph the origin of the *Xenopus* data (a diverse array of experimental data deposited in Xenbase) vs. the transcriptomes of mantellids determined by ourselves.

Overall, this comment also relates to one suggestion of reviewer #3. Following Reviewer 3’s more specific comments, we changed the title of our study to “Transcriptomic and macroevolutionary evidence for phenotypic uncoupling between frog life history phases” in order to better express the novelty of our approach in collecting the relevant evidence for an outcome that might not be so surprising for experts.

There are several wordings and other grammatical usages that might be improved for readability, e. g. “were found” rather than “turned out” (line 45)

Response: Done. “turned out” has been replaced with “were found” (line 45),

“composed of” rather than “joining” (line 79)

Response: Done. “joining” has been replaced with “composed of”

“such that” rather than “until” (line 88)

Response: Done. “until” has been replaced with “such that”

And “tadpoles did NOT serve” Harris anything (line 82); he (and Falconer) used data on tadpoles...

Response: Thank you for the suggestion. Replaced with “Harris, building on Falconer, used data on tadpoles to propose models...”

..and many others. There are also some word-order issues, but a quick copy-edit will take care of all of these minor points, if the authors wish.

Response: We have asked a colleague who is native-English speaker and knowledgeable in style and grammar of English language to copyedit the manuscript. As a result, multiple small improvements of wording have been applied.

All in all, very nice work!

Response: Thank you for the positive evaluation!

Reviewer #2 (Remarks to the Author):

Writing: drop all the directional words: though, indeed, likewise, however, etc. etc. Let the reader decide! - - I dislike the use of parenthetical words placed within a sentence! - - some wording needs attention = “phases are in Mantellidae” – the whole manuscript would benefit from a strong edit - -

Response: We have asked a colleague who is native-English speaker and knowledgeable in style and grammar of English language to copyedit the manuscript. As a result, multiple small improvements of wording have been applied, including the modifications requested by the reviewer.

Subject: I could not follow all the details of your analyses, but I found the results and discussion very exciting. The process you discuss is probably wide spread in the frog/tadpole sphere, and I hope you pursue other avenues: other taxa, phenotypic plasticity, other reproductive modes, etc.

Response: Thank you for the encouragement. We will pursue this subject further in the future. Most interestingly will be a comparison with other vertebrates having larval stages, e.g., salamanders, lungfish or bichir which all have larval stages more similar to the adults and where one could argue why uncoupling has not evolved to the same degree as it did in frogs.

Reviewer #3 (Remarks to the Author):

Wollenberg Valero et al. present a well designed and well conducted study on the independence of morphological evolution in tadpoles and their adult stage in frogs. The paper analyzes the (mostly) independent morphological evolution of larvae and adults in a substantial sample of Madagascan mantellid frogs, based on transgenomic and morphological data. An early attempt addressing adult morphology in comparison to larval morphological diversity in frogs was published by Bossuyt and Milinkovitch (2000), but the present study goes far beyond. To the best of my knowledge no other study in amphibians has addressed the question of independent larval/adult evolution in frogs in this depth and clarity. I am confident that this work will not only appeal to frog people. Researchers working on animals with complex life cycles

(e.g., insects) in general will be interested to read this work. Furthermore, it has relevance to other ongoing projects that try to understand the evolution of Madagascar fauna. In sum, I think this is an important paper that will have an influence on the thinking in the field.

Wollenberg Valero et al. presented a large dataset, including transgenomic data of two taxa and morphological data for more than 100 species. Their hypothesis building and argumentation is clear and the evidence supports the conclusions strongly. The supplementary materials cover all aspects of the analysis and all methods applied in reasonable depth. The figures provided are well done and give sufficient (and necessary) visual support to the ideas expressed in writing. The text is well structured.

Reading the text, I had only a few minor issues and I would like to share these thoughts, so the authors may consider them for improving this work:

I am not sure if the current title is the best choice. Anyone working on animals with complex life-cycle will actually expect that the life history phases ARE uncoupled, otherwise it would not make much sense to have a complex life cycle in the first place. The authors acknowledge that fact themselves (lines 321-323). In other words, the title seems to highlight the obvious rather than the unexpected new. The merits of this work are, in my opinion, in the combination of transgenomics and adaptive radiation/phenotypic evolution analyses in order to present sound and strong evidence for the common assumption of uncoupling. Maybe the authors are willing to reconsider that title and include some more key words referring to the major cornerstones of this work. It is a matter of taste, but I think the current title does not quite nail it.

Response: Good point. Following this suggestion, we have changed the title to “Transcriptomic and macroevolutionary evidence for phenotypic uncoupling in frog life history phases”. We hope that by doing this, we can now convey that the novelty of the paper lies in the type of evidence combined to solve a basic question about life-history evolution (or alternatively, to prove an outcome that might be expected but has so far never been empirically tested despite its important implications) As one point to mention, we think the reviewer has made a small terminological mistake when writing about “transgenomic” data. We have checked this term and did not find it applied to the kind of analyses we have carried out, and we therefore assume he has meant to say “transcriptomic”, which is the term we have therefore applied to changes in the title and text.

Technically sound data: I need to point out to the editors that I am familiar with some but not all methods applied in this study. Generally, I have the impression that the explanations in the Supplements have an appropriate level of explanation to offer the expert reader good depth of detail to reconstruct what had been done in this work. That said, some minor questions arose during my reading:

The morphological characters are presented in tables. The quality of character documentation is very mixed. Although I have plenty of experience in tadpoles I easily bumped into character definitions that I simply do not understand (for example, 117: Lateral space?; 116: classify all possible color patterns in only three states?; what were the landmark point for measurements; what is “body length axis” , etc., etc.). Obviously character definition in morphology can be very difficult if meant to be reproducible. Certainly, this submission is not necessarily the place to present a lengthy documentation of morphological character states to exclude all ambiguity.

Therefore, I am NOT asking for a revision of those tables(!), however, one question might be in place: Given the ambiguity or fuzzy definition of some morphological characters, does this ambiguity hold the potential to change the conclusions of the paper if morphological characters were handled differently (for example, five subjective color pattern [116] states rather than three?). I'd like to see the authors consider that and comment on it with a line in an appropriate section of the text.

Response: We are grateful for the reviewer to suggest improvement of our character state descriptions, which we now understand to have contained a number of typos and inconsistencies.

To correct these embarrassing shortcomings, we have spent a considerable amount of time to improve the documentation of characters and character states. We have completely reworked the supplementary tables S4 and S5 to more accurately describe how characters were coded, and to show distinction of the different character states. For clarification, we also now included a figure showing the landmarks of all tadpole measurements, as probably readers would be less familiar with these as with standard measurements of adult frogs (new supplementary figure S9).

We furthermore were intrigued by the reviewer's question on whether character coding might have affected our results. This is a very valid question, and it led us to reconsider a number of points.

First, we realized that we had not tested whether the larger number of characters of tadpoles (117 vs. 43 in adults) might have affected the MDS variable in a way that absolute values would be larger than in the analysis of tadpole characters, thereby artificially leading to higher rates of morphological change in tadpoles. We calculated 10 jackknife pseudoreplicates of the tadpole character set, leaving only 43 characters (the same number as in the adult character matrix), and for each of these replicates we calculated a 1-dimensional MDS variable. The resulting values were strongly correlated with the variable from the 117-character matrix, and especially, within the same range of values (-1.5 to 1.5). While this is

exactly the expectation, it is good to see it empirically confirmed despite the MDS data transformations involved. Hence, the higher evolutionary rates of tadpoles are a "true" phenomenon cannot be explained by potential data transformation artifacts arising from a higher number of tadpole characters scored.

Second, the same resampled data sets also served us to verify whether specific character coding methods might have strongly influenced or biased our results. We found that the values of the MDS variables resulting from the jackknifed character sets were strongly correlated to the original full character set. As from each of the jackknifed data sets almost 2/3 of the characters were excluded, the encountered strong correlation suggests that single characters of equivocal coding are very unlikely to strongly influence the downstream analyses (because these characters would with high probability be excluded in at least some of the dataset replicates).

We also repeated the rate analysis with the R-package *auteur* for one of these resampled data sets. We found that the obtained rates still were on average distinctly lower than those obtained from the adult data set, thus confirming that the number of characters had no influence on our conclusion of faster rates of morphological evolution in tadpoles.

To keep the manuscript short, we have refrained from reporting these additional analysis in the main manuscript, but we have added a sentence to Methods and also to Supplementary Methods (L. 190ff) stating that we performed exploratory analyses with different character coding schemes and character jackknifing to verify that neither the number of characters nor the coding scheme itself has influenced our main conclusions (i.e., faster rates in tadpoles, and uncoupling of rates among tadpoles and adults). And the results of the rate calculation are reported in a new Supplementary Table S9.

Please also see the second next response below in which we elaborate in additional detail on character coding issues.

The morphological datasets were subject to maximum parsimony analyses "to understand how phylogenetically informative the morphological data are" (Supplement page 3). The connection between the morphological parsimony analysis (line 178; Figure S4) to the subsequent steps of analyses have not become fully clear to me, because there is also the molecular tree (Wollenberg et al 2011) that was used as backbone, for example, to reconstruct morphological variation under a Brownian Motion model to determine rates of phenotypic evolution... In other words: There is the Wollenberg-2011 backbone molecular tree and there is the morphological parsimony analysis and the resulting morphology based tree (Fig. S4) and I am not sure why the latter is needed. Does the morphological parsimony tree feed directly into any of the subsequent analyses? Either it is obsolete, or it is essential and I missed some connections. A clarification will be welcome.

Response: We are grateful to the reviewer for these comments which led us to reconsider one topic that we also had discussed among the authors previously. In fact neither of the morphology-based trees (from supplementary figures S4) were used in any of the downstream analysis. Only the molecular tree was used for running "auteur", "Bayes Multistate" and "surface". We calculated the morphological, MP-based trees in an early stage of the research project and this took quite some effort, which is why we had decided to present them as Supplementary Material, although they are largely disconnected from the main analyses of the paper and have no relevance for our main conclusions.

However, from the comment of the reviewer we understand that keeping these trees does more harm than good by causing confusion about which tree was used for which analysis. We have therefore decided to completely remove them (Fig. S4 and Table S8 deleted), and have also removed the respective section of Supplementary methods, and any mention of this part of the analysis from the main manuscript.

If the parsimony analysis of morphological features directly feeds into subsequent analysis, then more questions follow. The morphological datasets contained phylogenetic signal (line 180). Yet, there are one has to ask about the process of transforming the many morphometric, quantitative and potentially continuous character states into categorial [sic.] (better: discrete). In most characters the metric intervals between character states were set equal, in others the intervals of values are unequal. I could not find a rationale for these different data conversions. It could be that the prior knowledge (bias?) of the authors may have shaped the decision of where to set the boundaries. Furthermore, the authors preferred coding of many characters as multi-state characters (State 0: absent, State 1: X; State 2: Y; for example, adult Char 1) that could alternatively be coded as two characters (absent/present; X/Y). These two ways of coding influence the reconstruction of apomorphies at nodes. Presence of a structure (in the example, presence of the femoral gland as an organ) cannot be reconstructed as synapomorphy at any node if coded as multi-state character (only the two subforms of the femoral glands). I wonder if these two different ways of handling character coding would have any effect on subsequent analyses of phenotypic evolution.

Response: See also the responses to the previous and subsequent comments of this reviewer. The

phylogenetic analysis based on the maximum parsimony optimality criterion has been removed from the manuscript as it was not relevant to the core study results presented herein.

Thus the question that remains is, has character coding possibly affected other downstream analyses? To test this, we applied the following strategy. First, we pruned the tadpole character data set to retain only a set of 40 characters that we considered representing well morphological variation of these tadpoles, i.e., including characters related to body form, relative length and shape of tail and tail muscles, size, position and form of oral disc, number, size and configuration of keratodonts and oral papillae, and color pattern. We verified that the MDS variable derived from this subset of characters was very strongly correlated with that from the full set of characters ($r=0.83$). We then re-coded the 40 characters into purely binary states as suggested by the reviewer, i.e., in some cases we separated a character into 2-4 binary characters, in other cases we simplified coding and merged a rare character state with morphologically closest common state. The result was a matrix of 44 characters, and the resulting MDS variable was still strongly correlated to that derived from the original data set ($r=0.51$), with a distinctly higher r value than the correlation between the original data sets for adults and tadpoles ($r=0.27$).

We then ran "auteur" for the MDS variable derived from this recoded matrix. We found that (1) the obtained rates were still distinctly lower than those derived from the adult data sets, thus demonstrating that independently from the coding scheme, tadpoles have a faster morphological evolution in mantellids. We also found that (2) the tadpole rates obtained from the recoded (as well as the resampled) data sets were correlated across branches with rates from the original tadpole data set (117 characters) but not with the adult data set. This demonstrates that independently of the coding scheme, rates of morphological evolution of tadpoles and adults are phylogenetically uncoupled in mantellids.

We have added the average rates of the different (new) data sets in a new Supplementary Table S9, and the correlation results to a new Supplementary Table S10, and are referring to these two tables in the Methods section of our main manuscript and in Supplementary Methods L. 190ff..

Down-weighting characters that are variable intra-specifically or may be affected by preservation artifacts may sound reasonable at first. However, what is the justification of the amount of down-weighting? The authors cite Goloboff et al. (2008) and mention "upweight % 8 downweight 4%" in the supplements. Why were these figures chosen and not others? Is there a justification for these figures, why not more/less? As a reference point and for transparency: would it change the results and conclusions if all characters were weighted equally?

Literature on character weighting: Farris 1990, Campbell and Frost 1993, Wiens 1995, 1998.

Response: We agree that character weighting is a tricky issue and definitely would have required more explanation and discussion than we had given it in this paper where the parsimony analyses were just reported in supplementary materials as a by-product of the study, not directly feeding into the subsequent analysis. Given the low relevance of the parsimony phylogenetic analyses for the study, we have now completely removed this part (trees in Fig. S4) from the manuscript. Hence, no analyses with character weighting remain in the manuscript.

I am not familiar with the concept of "selective phenotypic optima". When it is first mentioned in the text [line 212] no explanation or citation is given. The explanation in the Supplement section (page 4) did not quite clarify it for me. Sorry for my ignorance, but with respect to the broad readership and the frequent use of phenotypic optima in the text and central role of the term/concept, this work would benefit from making the concept more accessible for the reader.

Response: We agree with the reviewer that the concept of phenotypic optima needs to be explained better. The concept has been developed by Hansen, an analysis method has been developed in Ingram & Mahler 2013, and previously served to analyze anole (Mahler), and frog (Moen et al.) morphological evolution. Moen et al. (2016) write:

The estimated adaptive "optimum" in OU models is a statistical concept, not based on biomechanical or selection studies of what phenotype would function best in a given environment. Instead, it reflects a single phenotype toward which individual lineages evolve (Hansen 1997, 2012). Each lineage can have its own optimal phenotype due to its idiosyncratic evolutionary history and constraints, but the adaptive optimum estimated in the models (also called the primary optimum; Hansen 2012) is assumed to reflect selection due to a common factor shared by a set of species (in our case, those sharing a given microhabitat). It can take time to overcome constraints (e.g., genetic correlations, pleiotropy) and the impact of past environments on the phenotype. Both of these factors can cause species' phenotypes to differ from the adaptive optimum. Here, we

introduce a method to decompose the variation in species' phenotypes around their inferred adaptive optimum. Systematic deviation from the current optimum (toward the optima of ancestral environments) would indicate that historical factors have prevented species from reaching the same, convergent adaptive optimum. In contrast, random deviations around the optimum would suggest that history is unimportant.

A respective sentence has been added to the introduction L 212ff: "...Referring to Hansen (1997, 2012), Moen et al. (2016) define a phenotypic, also called selective or adaptive optimum, as a hypothetical phenotype towards which one or more lineages can evolve ...". We also added to the next sentence that the method we used and that are described in detail in the supplementary materials, follow Ingram and Mahler 2013.

Recommendation: Accept with revisions
A. Haas

REVIEWERS' COMMENTS:

Reviewer #1 (Remarks to the Author):

I am quite positively impressed with the thoughtful, thorough revision that the authors have done. My concerns are certainly dealt with well. I have read thoroughly through the authors' responses to the three reviewers' comments, and then the revised ms itself. I find all the small editorial/grammatical changes effectively done, the title change highly appropriate, and the more substantial issues dealt with in a manner that has provided considerable new clarity to the manuscript. Obviously, serious thought was given to the revision, and it has been effective. So often, authors respond by simply removing passages that were found to be unclear. These authors have done a real revision, and the few removals are explained and justified, and the material will be included in future publications as appropriate.

I expect that the ms will be a notable contribution advancing theory and practice in evolutionary biology, life history analysis, and the phylogenetic basis for evaluating such patterns.

Excellent job!

Reviewer #3 (Remarks to the Author):

Thanks very much for asking again for my opinion about Wollenbegr Valero et al. I went over the documents again and conclude that the authors have taken great care to respond to all of my concerns. The title has been changed and has, in my opinion, more punch and specificity now. In other cases (such as the the morphology tree) the authors preferred to excise blocks of information rather than justifying them by lengthy explanations. I think this was a wise way of handling the problems without loss to the major message and merit of this work. By doing so, the work gained additional clarity. The supplement section is giving good and precise information about the methods applied and analysis parameters chosen. As I am quite happy with the current version of the manuscript, I do not consider it important to go through all points of the rebuttal letter again. Let me just say that in sum, I am fully satisfied with the authors careful and thoughtful revisions and that I have no further objections. I congratulate the authors and wish them good luck with this work!
Alexander Haas